# Enhancement of Scaffold In Vivo Biodegradability for Bone Regeneration Using P28 Peptide Formulations

**DOI:** 10.3390/ph16060876

**Published:** 2023-06-13

**Authors:** Farah Alwani Azaman, Margaret E. Brennan Fournet, Suzina Sheikh Ab Hamid, Muhamad Syahrul Fitri Zawawi, Valdemiro Amaro da Silva Junior, Declan M. Devine

**Affiliations:** 1PRISM Research Institute, Technological University of the Shannon (TUS), N37 HD68 Athlone, Ireland; f.alwani@research.ait.ie (F.A.A.);; 2Tissue Bank, School of Medical Sciences, Health Campus, Universiti Sains Malaysia (USM), 16150 Kota Bharu, Malaysia; 3Departamento de Medicina Veterinária, Universidade Federal Rural de Pernambuco, Recife 52171-900, Brazil

**Keywords:** bone tissue engineering, scaffold degradation, chitosan, osteogenesis, bone regeneration

## Abstract

The field of bone tissue engineering has shown a great variety of bone graft substitute materials under development to date, with the aim to reconstruct new bone tissue while maintaining characteristics close to the native bone. Currently, insufficient scaffold degradation remains the critical limitation for the success of tailoring the bone formation turnover rate. This study examines novel scaffold formulations to improve the degradation rate in vivo, utilising chitosan (CS), hydroxyapatite (HAp) and fluorapatite (FAp) at different ratios. Previously, the P28 peptide was reported to present similar, if not better performance in new bone production to its native protein, bone morphogenetic protein-2 (BMP-2), in promoting osteogenesis in vivo. Therefore, various P28 concentrations were incorporated into the CS/HAp/FAp scaffolds for implantation in vivo. H&E staining shows minimal scaffold traces in most of the defects induced after eight weeks, showing the enhanced biodegradability of the scaffolds in vivo. The HE stain highlighted the thickened periosteum indicating a new bone formation in the scaffolds, where CS/HAp/FAp/P28 75 µg and CS/HAp/FAp/P28 150 µg showed the cortical and trabecular thickening. CS/HAp/FAp 1:1 P28 150 µg scaffolds showed a higher intensity of calcein green label with the absence of xylenol orange label, which indicates that mineralisation and remodelling was not ongoing four days prior to sacrifice. Conversely, double labelling was observed in the CS/HAp/FAp 1:1 P28 25 µg and CS/HAp/FAp/P28 75 µg, which indicates continued mineralisation at days ten and four prior to sacrifice. Based on the HE and fluorochrome label, CS/HAp/FAp 1:1 with P28 peptides presented a consistent positive osteoinduction following the implantation in the femoral condyle defects. These results show the ability of this tailored formulation to improve the scaffold degradation for bone regeneration and present a cost-effective alternative to BMP-2.

## 1. Introduction

Biodegradable scaffolds for bone tissue engineering are of great interest to researchers due to their promising characteristics and performance in mimicking the extracellular matrices in promoting natural bone healing [1,2,3]. These engineered scaffolds are great alternatives to the natural-sourced treatments available, such as bone grafts, despite their gold-standard properties, due to the multiple surgical interventions required (autografts) as well as the existing risk of immune rejection (allografts) [4,5,6].

A common approach to the development of biodegradable scaffolds is to initially culture osteogenic cells containing growth factors on 3D scaffolds prior to implantation, as suggested by the diamond concept, to simulate the physiological conditions [7,8,9,10]. However, the additional cellular harvesting and culturing on the scaffolds needed prior to use are time-consuming, thus leading studies towards acellular osteoconductive and osteoinductive scaffolds with osteogenic growth factors [11,12,13]. The current study combines biology and engineering principles to develop viable substitutes to restore and maintain the function of human bone tissue. It aimed to enhance the mechanical and biodegradable properties of bone scaffolds by modifying the crosslinking reaction to avoid the burst release of the growth factors. The fabricated biodegradable scaffold will release growth factors while degrading and then be secreted from the body naturally after it completes its function. Therefore, an ideal scaffold should be able to degrade in a gradational way for an expected period in order to be replaced by newly formed bone tissue from the bonded cells, known as the osteotransduction process [14]. This degradation will result in the breakdown of the scaffold and the resorption of the protein, such as bone morphogenetic protein-2 (BMP-2), incorporated in the scaffold to the targeted location.

The current market leader for a Food and Drug Administration (FDA) approved growth factor delivery system is the recombinant human BMP-2 (rhBMP-2), known under the trade name INFUSE^®^, with an absorbable collagen sponge carrier to be used as a bone graft substitute in treating open tibial fractures. This product is in line with also-FDA-approved AUGMENT^®^ by Wright Medical and i-Factor by Cerapedics [15,16,17,18]. However, this INFUSE^®^ treatment has had limited success in the treatment of non-union healing, and this product is associated with several documented complications, such as ectopic calcification and bone formation, as well as transient bioactivity, which is the off-target reaction [19,20,21,22]. INFUSE^®^ has also faced an FDA warning following reports of severe dysphagia due to inflammation, as well as the increased loss of life.

An alternative to the use of BMP-2 is the use of osteogenic peptides derived from BMP. Peptides are advantageous due to their small relative molecular weight, known physiological effect, and lower cost demands [23]. There are two sites of interest in the complex structure of the BMP-2 dimer, known as wrist and knuckle epitope regions. Extensive reports have proposed the ability of peptides derived from this knuckle region to induce osteogenesis, thus enabling it to substitute the full length of rhBMP-2 [24]. Initially, a synthetic peptide, P4, synthesised from the knuckle epitope of BMP-2 (73–92), was reported to increase the alkaline phosphatase (ALP) and osteocalcin activity to the highest levels in the murine multipotent mesenchymal cell line (C3H10T1/2), compared to the other BMP-2 derived peptides [25,26]. This BMP-2 knuckle epitope-derived peptide (P4) has also been shown to increase osteopontin and mineral deposition in clonally derived murine mesenchymal stem cells (7F2) [27]. Subsequently, another short BMP-2-related peptide called P24, with a molecular weight of 2630.88 g/mol, was then synthesised [28]. This peptide consists of chemically stable small molecules and a linear structure as a biologically active site, and is believed to promote bone marrow stromal cell adhesion, enhance ectopic osteogenesis, and repair critical-sized rabbit bone defects [29].

Following the P24 synthesis, Cui et al. [30] have improved the work mentioned above by synthesising a longer BMP-2 dimer-knuckle epitope-derived peptide chain called P28 (S^[PO4]^DDDDDDDKIPKASSVPTELSAISTLYL) [31]. This P28 peptide possesses several significant properties compared to BMP-2, in terms of its smaller relative molecular weight and better chemical stability, that can improve its biological effects. The most impressive feature of P28 in this bone tissue engineering field is its repetitive amino acid sequences, with high bonding ability towards calcium phosphate materials. This feature can lead to an extended release with higher delivery specificity to the intended site, thus representing the potential for use in bone substitute research [23]. Moreover, the biomimetic feature of peptides in retaining the osteogenic features of the larger proteins offers greater control over cellular interactions. The shorter chains of peptides are advantageous in overcoming the steric effects, folding, immunogenicity and susceptibility to degradation problems of the larger proteins, thus leading to better signalling and binding domain availability for the required cellular interactions [32].

To date, the scaffolds’ slow degradation profile has been found to impede in vivo bone formation using growth factor therapies [7,18,20]. Hence, despite the great potential of the osteogenic peptide P28, there is a need to design a fully functional carrier to ensure it is delivered to the injury site intact and retained in the defect until it has fulfilled its function. The performance of bone substitute materials can be improved by tuning the scaffold composition and fabrication method. This work employed the revised P28 delivery systems, which were designed to have faster degradation than previous formulations [7] by incorporating the combinations of hydroxyapatite and fluorapatite ceramics into the chitosan composite. In addition, the osteogenicity of different P28 concentrations was tested in vivo based on the previous work [20]. Therefore, an increased P28 content (75 and 150 µg) was also tested in CS/HAp/FAp scaffolds in order to investigate the best osteogenicity effect on the rat femoral condyle defect model utilised in this work.

## 2. Results and Discussion

### 2.1. Animal Husbandry

The weights of the animals were monitored from the acclimatisation period until sacrifice. The weight was increased during the acclimatisation period until the implantation day. Following implantation, the weight decreased for about a week, which was a sign of pain and discomfort in post-operative procedures and the effects of anaesthesia and analgesia that led to decreasing appetite or lameness [33]. In addition, several incidents of broken sutures had exposed the wound, increasing the risk of pain and infection to the wound, while the re-suturing procedures required anaesthesia and analgesia that further caused the reduction in catching food. However, the ambulation and weight loss were within set limits commonly used (20% weight loss post-surgery) [34], and the weight of all animals began to increase again after 7–10 days, indicating good tolerance of the selected model.

### 2.2. Histopathological Evaluation

All femoral condyles with the implanted scaffolds from the six formulations (*n* = 3) were harvested, and the local macroscopic condition was evaluated (Table 1) according to the modified scoring table from Rudert et al. [35]. Scaffold-only controls (negative) and Infuse^®^ (positive) were previously reported [20]. To avoid the unnecessary duplication of data and animals, these controls were not repeated in this work. While empty controls were not used, the 3 mm defect in the rat femoral defect was not considered a critical size, and as such is expected to heal.

Several remarks were observed macroscopically, where most defects were closed with a layer of transparent tissue. While new bone growth was seen around the defects in the presence of P28 (12% HW CS/HAp/FAp 1:1/P28 25 µg), some of them showed scaffold residues, and more scaffold residues were seen in scaffolds formulated with higher HAp content (12% HW CS/HAp/FAp 1:0.75). In addition, there was a possibility of ectopic growth seen in scaffolds with the highest content of P28 (12% HW CS/HAp/FAp 1:1/P28 150 µg). On the other hand, an incidence of drilling burr slippage from the site of interest had occurred on the right condyle of R6, producing a defect on the bone shaft instead of the condyle. Therefore, the result from this condyle was completely discounted. Altogether, the Kruskal–Wallis test showed *p* > 0.05 for the investigated criteria; however, the chi-square approximation from this test may not be accurate since the sample size was less than five.

### 2.3. Histological Analysis

#### Hematoxylin-Eosin Staining

Histological analysis through HE staining was conducted to evaluate the performance of the implanted scaffolds and presented according to each formulation (Figure 1). Interestingly, the scaffold traces were observed to be minimal in most of the defects induced. This observation might validate that the combination of HAp and FAp in the chitosan-based delivery system has increased the biodegradability of the scaffolds in vivo, although it was reported that FAp alone possesses a lower solubility in biological fluids compared to HAp [36,37]. The behaviour observed was also in line with in vitro biodegradability results reported previously [7], where the CS/FAp scaffold lost integrity during handling as early as week two following submersion in simulated body fluid.

From the review of the HE-stained sections, the projection of fibrous tissue containing central ossification separated the spongy bone tissue areas in the implanted defect region (Figure 1A). Cortical bone thickening was observed in the bone implanted with 12% HW CS/HAp/FAp 1:1 scaffold (Figure 1B), which could result from the osteointegration of the composite. Moreover, the thickening of the periosteum near the defect area with an external surface irregularity of the compact bone, as well as osteointegration with the implant below the periosteum, was observed for 12% HW CS/HAp/FAp 1:0.75 (Figure 1C). Similar periosteum thickening could also be seen in the defect implanted with 12% HW CS/HAp/FAp 1:1 P28 25 µg (Figure 1D). On the other hand, the cortical and trabecular bone thickening (arrow) in the defect region of 12% HW CS/HAp/FAp 1:1/P28 75 µg might be due to osteointegration and induction from the implantation of the composite (Figure 1E). Compact bone tissue with two different densities (A,B), as well as the thickened periosteum (arrow), was present in the defects with 12% HW CS/HAp/FAp 1:1 P28 150 µg, which might be derived from the osteoinductive activity of the composite with the highest peptide content (Figure 1F).

Enhanced bone formation resulting from the use of tissue engineering scaffolds incorporating the P28 peptide has also been reported in the literature. In the initial studies reporting the use of P28, the silicone/hydroxyapatite (Si/HAp) scaffold was loaded with P28 peptide and implanted in rat calvarial defects [30]. The defects showed that the Si/HAp/P28 scaffold promoted bone recovery to a similar degree as the Si/HAp/rhBMP-2 scaffold. No new bone formation occurred in the empty defects (control) 6 weeks after surgery, although fibrous tissues were observed in the defect area, and the recovery did not improve, even 12 weeks post-implantation. Similarly, in the later study by Chao et al. [31], the control in a canine (dogs) defect model that was implanted with hydroxyapatite/ß-tricalcium phosphate/collagen (HAp/TCP/Col) scaffold alone revealed limited bone regeneration with noticeable HAp/TCP particles at four and eight weeks compared to the defects implanted with HAp/TCP/Col/P28, thus showing the positive bone regenerating capacity of this BMP-2 derived peptide.

Earlier studies utilised a high concentration of P28 peptide in order to bind the peptide onto the scaffolds through the physiosorb method to evaluate the osteogenic induction of bone defects in the presence of this P28 peptide [23,38,39]. However, no previous reports were found to investigate the efficacy of the P28 scaffolds in different mass content towards the osteogenic induction of bone defects using UV crosslinking in CS/HAp/FAp scaffolds. Therefore, the strongest new bone formation achieved with 150 µg P28 in this work could be a potential reference for future work involving P28 peptides. This early histological evaluation of the P28 peptide’s osteogenicity in an in vivo setting offers convincing proof of the compound’s potential in bone tissue engineering and regenerative medicine. The findings of this study indicate that the P28 peptide may increase the quality of newly produced bone tissue and stimulate bone growth, but more studies are required to validate its efficacy and safety in people.

### 2.4. Fluorochrome Labelling Analyses

Fluorescent labelling was evaluated as the dynamic measure of bone formation that corresponds to mineralisation speed, time, and location, as well as the direction of the mineralisation [40,41]. Calcein green and xylenol orange fluorochromes were injected in vivo ten and four days before sacrifice, respectively. The fluorochrome labels work by chelating the mineralised front during the bone mineralisation process at the time of injection, which enables the evaluation of the dynamic bone formation by measuring the distance between fluorochromes labels [42]. It has also been documented that fluorochrome labelling can be used to evaluate the bone mineralisation rate, remodelling, and the signs of toxicity in implants on the bone in vivo [14].

From the analysis of fluorochrome labels (Figure 2), it was found that the presence of the XO label was observed in the condyle implanted with the 12% LW CS/HAp/FAp 1:1, indicating that mineralisation was still ongoing four days before sacrifice. However, the absence of CG may indicate that this mineralisation was in the form of remodelling bone formed earlier in the healing process. In contrast, the condyle implanted with 12% HW CS/HAp/FAp 1:1 showed the presence of double labelling, where a positive label was seen in both CG and XO. This double labelling indicated ongoing bony deposition on the injection days. In addition, a single CG label appeared in the condyle implanted with 12% HW CS/HAp/FAp 1:0.75 and 12% HW CS/HAp/FAp 1:1 P28 150 µg, indicating that bone formation had occurred ten days prior to sacrifice. Since the XO label was absent, this indicates that no bone formation occurred 4 days prior to sacrifice in this single condyle. This observation could indicate that bone remodelling had finished at that time, prior to XO administration [20,41]. The samples with 12% HW CS/HAp/FAp 1:1 P28 25 µg and 12% HW CS/HAp/FAp 1:1 P28 75 µg shared a similar observation, whereby double labelling appeared in the condyles, indicating the ongoing mineralisation at ten days and four days before sacrifice.

Fluorochrome labels, such as CG and XO, bind to mineralised tissues, including both newly formed bone and bone-like ceramics. These labels can provide information about the dynamic process of mineralisation, regardless of whether it occurs in natural bone or synthetic ceramics, thus allowing for the evaluation of the mineralisation process in both materials [43]. However, the evaluations in this work were carried out considering the defect margin and not including the implanted composite scaffolds in order to distinguish between the two [40,44,45,46]. Since the accuracy and interpretation of the results obtained using fluorochrome labels can be influenced by various factors, including the specific labelling technique and the interaction between the labels and the materials being studied [47], a quantitative assessment of fluorochrome labels is recommended in order to achieve a definite comparison between samples, evaluating the mineral apposition rate (MAR) and also the bone formation rate (BFR). These measurements are proposed since the length of fluorescent labels and the distance between the labels are measurable parameters which can be used to evaluate the new bone turnover [48,49]. However, due to the small sample size and a lack of clear bone growth fronts in this work, these measurements were not possible.

## 3. Materials and Methods

Chitosan (high MW), hydroxyapatite, ethanol ≥ 99.8%, sodium fluoride ≥ 99%, Harris hematoxylin, eosin B and Entellan™ mounting medium were obtained from Sigma Aldrich ((Merck KGaA, Darmstadt, Germany). Sodium bicarbonate 99.5% was purchased from Acros Organics (Fisher Scientific UK Ltd., Loughborough, UK), poly(ethylene glycol) (600) dimethacrylate was obtained from Polysciences Inc. (Polysciences Europe GmbH, Germany) and benzophenone, 99% (A10739.30) was purchased from Alfa Aesar (Thermo Fisher (Kandel, Germany). P28 Peptide sequence >98% purity was synthesised by Pepmic (Pepmic Co. Ltd., Suzhou, China). Ketamine, 90 mg/kg (Narketan^®^-10, Vetoquinol UK Ltd., Towcester, UK), Xylazine, 5 mg/kg (Xylaxin^®^, Med Vet Biolinks Pvt. Ltd., Maharashtra, India), Lidocaine hydrochloride injection 2%, 4 mg/kg (Fresenius Kabi, Lake Zurich, IL, USA) and Isoflurane gas (Abbot Laboratory Ltd., Sittingbourne, UK) were used for general anaesthesia. Ethanol, povidone-iodine Betadine^®^ (Mundipharma Pharmaceuticals Sdn. Bhd., Petaling Jaya, Malaysia), chlorhexidine (Hibiscrub^®^, Manchester, UK), Opsite spray (Smith and Nephew, Hull, UK) and normal saline (RinsCap^®^, Ain Medicare, Malaysia) were also used. For double-fluorochrome injections, Calcein and xylenol orange tetrasodium salt were purchased from Merck, (Merck Life Science UK Limited Dorset, UK). Dorminal 20% (200 mg/mL pentobarbital sodium) was purchased from Alfasan International, Woerden, Netherlands. All materials were used as received.

The experimental procedures are summarised in Figure 3

### 3.1. Scaffold Preparation

Six different formulations with varied chitosan molecular weights, bioceramics ratios and P28 peptide concentrations were prepared as outlined in Table 2. 12% LW CS/Hap/Fap 1:1, 12% HW CS/Hap/Fap 1:1, 12% HW CS/Hap/Fap 1:0.75, 12% HW CS/Hap/Fap 1:1/P28 25 µg, 12% HW CS/Hap/Fap 1:1/P28 75 µg and 12% HW CS/Hap/Fap 1:1/P28 150 µg. The compositions of these scaffolds are outlined (Table 2), and were crosslinked using a UV curing system (Dr. Gröbel UV-Electronik GmbH, Opsytec Dr. Gröbel, Ettlingen, Germany) under 20 UV lamps with a spectral range between 315 and 400 nm and at the average intensity of 10–13.5 mW cm^2^ for 40 min. All the samples were flipped over mid-process.

### 3.2. Animal Housing and Husbandry

The animal study was conducted in the Animal Research and Service Centre, Universiti Sains Malaysia (ARASC USM), following animal ethical use approval (USM/IACUC/2020/(122)(1048)). Nine Sprague Dawley rats aged 11 weeks with initial weights ranging from 300 g to 400 g as received were used. Animals were housed in adapted installations (air-conditioned rooms with a temperature of 22 ± 3 °C and a 50–60% humidity level) to maintain them in an acclimatised environment and to avoid stress. Animals were followed up daily to detect any sign of stress or pain; in that case, analgesic treatment could be applied.

The animals were housed individually in tagged plastic boxes of standard dimensions and acclimatised for a minimum of seven days following the separation from their colonies in the breeding centre before the implantation. The general state of the animals was monitored daily. The artificial day/night light cycle was set to 12 h of light and 12 h of darkness. All animals had free access to water and were fed ad libitum with commercial chow daily. Cages were cleaned and changed weekly to prevent any unwanted infections at the surgical wound of the rats and to protect their health generally.

### 3.3. Femoral Condyle Defect Induction and Scaffold Implantation

The scaffold implantation was conducted using aseptic techniques. The implantation comprised 3 mm round defects on both sides of the femoral condyles, which were adapted and modified from work carried out by Klein et al. (2019) and Mohiuddin et al. (2019) [50,51]. All instruments and apparatus were set up in a cleanroom.

The procedure started with anaesthesia induction with an intraperitoneal injection of ketamine and xylazine. The doses of all drugs were calculated based on animal weight, which was measured just before the surgery using the formula below:$$\mathrm{Dosage}\;\left(\mathrm{mL}\right)=\frac{\mathrm{weight}\;\left(\mathrm{kg}\right)\times \mathrm{dose}\;\left(\frac{\mathrm{mg}}{\mathrm{kg}}\right)}{\mathrm{concentration}\;\left(\frac{\mathrm{mg}}{\mathrm{mL}}\right)}$$

Mixtures of oxygen/isoflurane (oxygen (0.4 L/min)/Isofluorane (1.5–2%) were given as anaesthesia maintenance throughout the surgery. The flanks of the animals were shaved by using an electronic hair shaver. Chlorhexidine was applied to the exposed skin, followed by povidone-iodine in an outward circular motion. A scalpel blade was used to make a firm incision on the skin into the muscle. The muscle mass was then dissected, exposing the bone surface of the femoral condyle. A 3 mm size defect was induced with a commercial micro drill equipped with a 3 mm bur (Figure 4). Saline irrigation was applied while drilling.

The chitosan-based scaffold was quickly inserted into the defect in a randomised manner (Table 3). Subsequently, the muscle was sutured with an absorbable suture, followed by suturing of the skin with the same absorbable suture. Iodine was reapplied and a transparent film dressing spray, Opsite spray (Smith&Nephew, Hull, UK), was applied as a final layer to protect the wounds. The procedures were repeated on the contralateral condyle. Dexamethasone, an anti-inflammatory drug (Dexavet 0.5%, Range Pharma, Malaysia), was injected intramuscularly before returning to the cage. The post-operative health condition, weights, wound healing and the behaviour of the animals were monitored twice daily for the first week, daily in the second week and every two or three days for the rest of the eight weeks study until the euthanasia procedure. A summary of the surgical procedures is outlined in Figure 5

### 3.4. Post-Operative Monitoring

The general post-operative conditions of the animals were monitored daily, and deeper monitoring in terms of body weight was recorded every two days in the early weeks post-implantation and every four days towards the end of the experiment. These monitoring procedures were carried out to observe whether the animals had reached the endpoints below, where removal from the study would be recommended:Weight loss >20% of the mean weight of rats;Severe lameness;Diarrhoea/blood in faecal material;Circling phenomenon;Severe necrosis at the implantation site;Persistent self-induced trauma five days after analgesic treatment as well as local and general treatment;Abnormal behaviour even in the presence of appropriate treatment (e.g., sign of pain following administration of analgesic).

### 3.5. Fluorescent Bone Labelling for Dynamic Bone Formation

All rats were subcutaneously injected with double-fluorochrome labelling: calcein green (CG) and xylenol orange (XO) (ten and four days, respectively) prior to sacrifice in order to highlight the calcification formed [40]. Calcein injection (10 mg/kg) was prepared by dissolving 0.1 g of calcein powder in a 2% sodium bicarbonate solution under sterile conditions [52,53,54]. The solvent was first prepared by dissolving 0.05 g sodium bicarbonate powder in 10 mL of 0.9% sterile saline solution, making the concentration 10 mg/mL. Xylenol injection (90 mg/kg) was prepared by weighing 0.2 g sodium bicarbonate and dissolving it in 10 mL of 0.9% sterile saline solution under sterile conditions. The solvent was poured into 1 g of xylenol orange tetrasodium salt and shaken until dissolved. The dosage for these double-fluorochrome injections was calculated based on the formula below and injected subcutaneously four days (calcein) and ten days (xylenol) prior to sacrifice:Dosage (mL) = (weight (kg) × dose (mg/kg))/(concentration (mg/mL))

### 3.6. Animal Euthanasia

Rats were sacrificed humanely via cardiac puncture under anaesthesia after eight weeks. The rats were anaesthetised using Dorminal 20% (200 mg/mL pentobarbital sodium) purchased from Alfasan International, Woerden, Netherlands, intraperitoneally (200 mg/kg) prior to cardiac puncture, which was carried out immediately after anaesthesia was induced.

Following euthanasia, the femurs of the rats were collected by incising the muscle until reaching the femoral head, and the ligaments holding the femoral head and the condyles were cut. The surrounding muscle was removed as much as possible. The femurs were then fixed in 10% neutral buffered formalin and cut using a hard tissue cutter, obtaining the implanted femoral condyles.

### 3.7. Macroscopic Histopathological Evaluation

A local macroscopic evaluation was carried out by observing the exposed femoral condyle implant sites. The observation focused on the defects’ visibility, colour, and texture through a macroscopic scoring system adapted and modified from Rudert et al. (2005) [35]. This scoring system (Table 4) was also applied in several other studies involving in vivo osteochondral experiments [55,56,57], and reviewed [58].

### 3.8. Histological Processing, Embedding and Cutting

Tissue samples were fixed in 10% neutral buffered formalin for two weeks prior to histological processing. Histological processing involved dehydration in ascending series of ethanol (70–100%), clearance in chloroform and infiltration with PMMA Technovit^®^ 7200 (Heraeus Kulzer GmbH, Wehrheim, Germany) in ascending concentrations (10–100% PMMA in ethanol), prior to embedding in Technovit^®^ 7200 hard resin with 1% benzoyl peroxide using blue light polymerisation, kept cool throughout the curing process.

Serial longitudinal sections of 100–300 µm thickness were made at the defect level to obtain the sagittal view of the defect using EXAKT diamond embedded saw equipment, and were polished using an EXAKT grinder machine to obtain 3–7 µm sections (EXAKT Advanced Technologies GmbH, Norderstedt, Germany). Subsequently, the staining procedures were performed using Hematoxylin-Eosin (HE) stain. Each stained slide was observed in a global view (×20 objective) using a brightfield microscope. In addition, unstained sections were directly observed under a fluorescent microscope to analyse the fluorochrome labels administered prior to the sacrifice.

### 3.9. Histological Staining (Hematoxylin/Eosin)

The staining procedures were conducted and evaluated by a histopathologist. The resin cuts were immersed in Harris hematoxylin dye for 50 s, after which they were washed in running water to remove excess dye. Subsequently, the same cuts were immersed in 1% eosin in alcohol for 180 s. They were then washed in 70% alcohol for 30 s and left on the bench to dry at room temperature for 24 h. After drying, the slide was mounted with Entellan™ and a coverslip prior to viewing under the light microscope [59].

### 3.10. Fluorescent Imaging

Fluorochrome labels were assessed on the non-stained sections using an Invitrogen EVOS M7000 fluorescent microscope with M7000 software (Life Technologies Corporation, Frederick, MD, USA). The labels were used to mark the bone formation on the injection days [45]. The fluorochrome labels were analysed using three light sources: GFP, RFP and transmitted light (brightfield). The GFP emission filter permitted fluorescent signals with wavelengths of 510 nm to 540 nm to pass through (CG), while the RFP emission filter allowed signals with 575 nm to 590 nm to pass through (XO), thus producing clean signals and images.

## 4. Conclusions

A preliminary in vivo trial was conducted bilaterally on nine male Sprague Dawley (SD) rats to evaluate the osteogenic potency of CS/Hap/FAP scaffolds loaded with P28 peptide. In this study, our aim was to provide evidence for a rationale of the effect of changing the ceramic content and changing the P28 concentration. As such, this study was designed to narrow down the variables for future experiments (identifying the best ratio and the best concentration of P28). Post-mortem observations indicate that most defects were observed to be closed with a layer of transparent tissue, while new bone growth was seen around the defects in the presence of P28 (12% HW CS/HAp/FAp 1:1/P28 25 µg). Additionally, there was a possibility of ectopic growth seen in scaffolds with the highest content of P28 (12% HW CS/HAp/FAp 1:1/P28 150 µg).

For histological assessments, unstained sections were assessed for the presence of fluorochrome labels, with HE staining used to assess the biological response to the scaffolds. The HE stain highlighted the thickened periosteum, indicating a new bone formation in the scaffolds, where CS/HAp/FAp/P28 75 µg and CS/HAp/FAp/P28 150 µg showed the cortical and trabecular thickening as a result of the implantation of composite in the region below the metaphysis. Subsequently, the CG and XO fluorochrome assessment indicated that CS/HAp/FAp 1:1 P28 150 µg scaffolds showed a high-intensity response to the calcein green label, indicating that mineralisation was ongoing 10 days prior to sacrifice. Similarly, the presence of double labelling, which is indicative of ongoing mineralisation, was observed at days ten and four prior to sacrifice in the CS/HAp/FAp 1:1/P28 25 µg and CS/HAp/FAp/P28 75 µg. Based on the HE and fluorochrome label, CS/HAp/FAp 1:1 with P28 (especially in the 12% HW CS/HAp/FAp 1:1/P28 25 μg scaffold) showed some indications of new bone growth around the defects, but further investigation is needed to confirm the consistent positive osteoinductive effects following the implantation in the femoral condyle defects. However, further investigation is required to rule out ectopic bone formation, especially in the highest P28 content. While partial progress has been made on the technical milestones, significant further work is required to fully understand the therapeutic potential of this P28-loaded CS/HAp/FAp scaffold. Additionally, further study is required to find the ideal dose and delivery technique in order to guarantee the P28 peptide’s safety and effectiveness in people.

## Figures and Tables

**Figure 1 pharmaceuticals-16-00876-f001:**
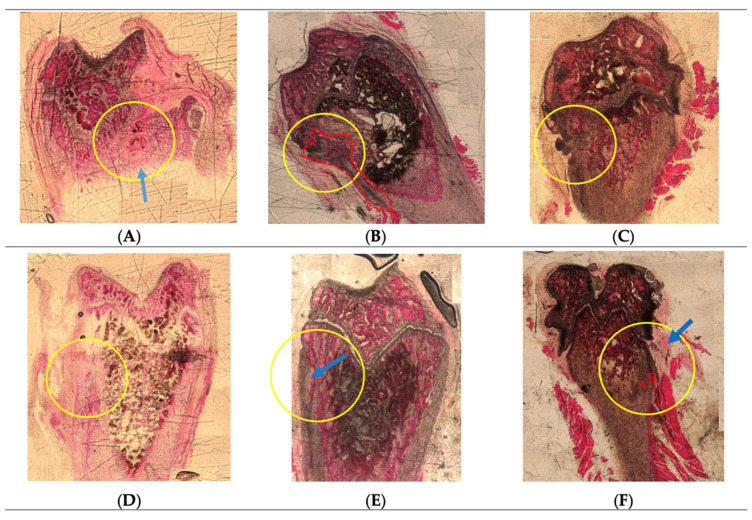
HE-stained slides of the implanted condyle defects. Yellow circles highlight the site of implanted defects. Blue arrows represent the newly formed bone structure where the composite was implanted. Red colouring (**A**,**B**) shows the two different densities of the compact bone tissue. (**A**) 12% LW CS/HAp/FAp 1:1. (**B**) 12% HW CS/HAp/FAp 1:1. (**C**) 12% HW CS/HAp/FAp 1:0.75. (**D**) 12% HW CS/HAp/FAp 1:1 P28 25 µg. (**E**) 12% HW CS/HAp/FAp 1:1 P28 75 µg. (**F**) 12% HW CS/HAp/FAp 1:1 P28 150 µg.

**Figure 2 pharmaceuticals-16-00876-f002:**
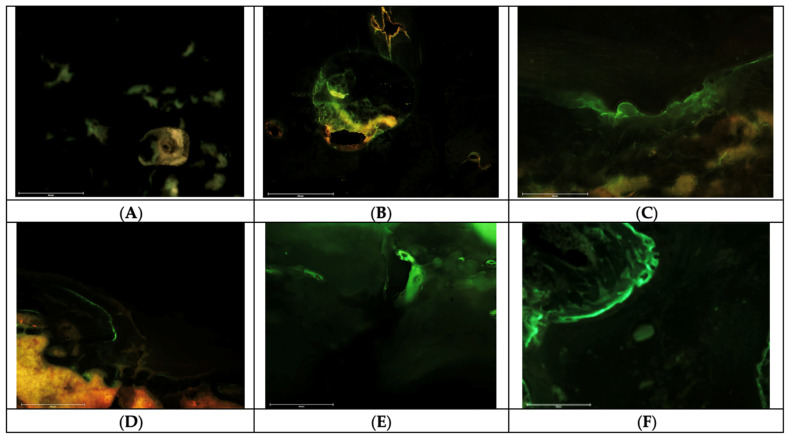
Fluorescent labelling imaging, observing the calcein green and xylenol orange labelling formed on the newly formed bone. (**A**) 12% LW CS/HAp/FAp 1:1. (**B**) 12% HW CS/HAp/FAp 1:1. (**C**) 12% HW CS/HAp/FAp 1:0.75. (**D**) 12% HW CS/HAp/FAp 1:1 P28 25 µg. (**E**) 12% HW CS/HAp/FAp 1:1 P28 75 µg. (**F**) 12% HW CS/HAp/FAp 1:1 P28 150 µg.

**Figure 3 pharmaceuticals-16-00876-f003:**
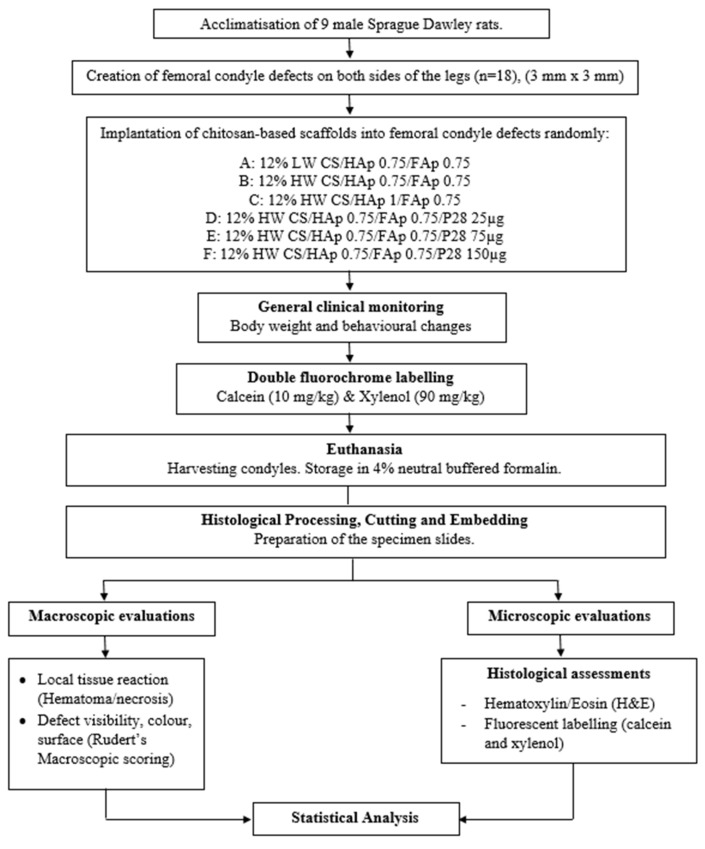
Summary of the in vivo experimental procedures.

**Figure 4 pharmaceuticals-16-00876-f004:**
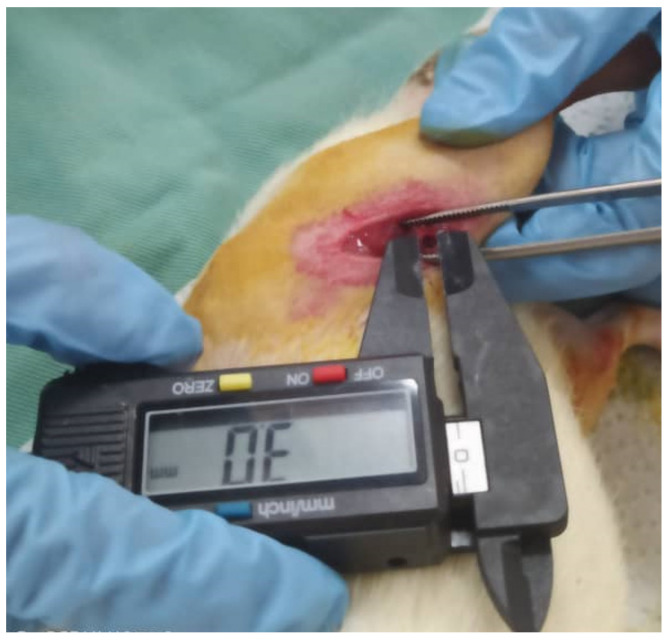
The 3 mm defect made on the femoral condyle of an SD rat.

**Figure 5 pharmaceuticals-16-00876-f005:**
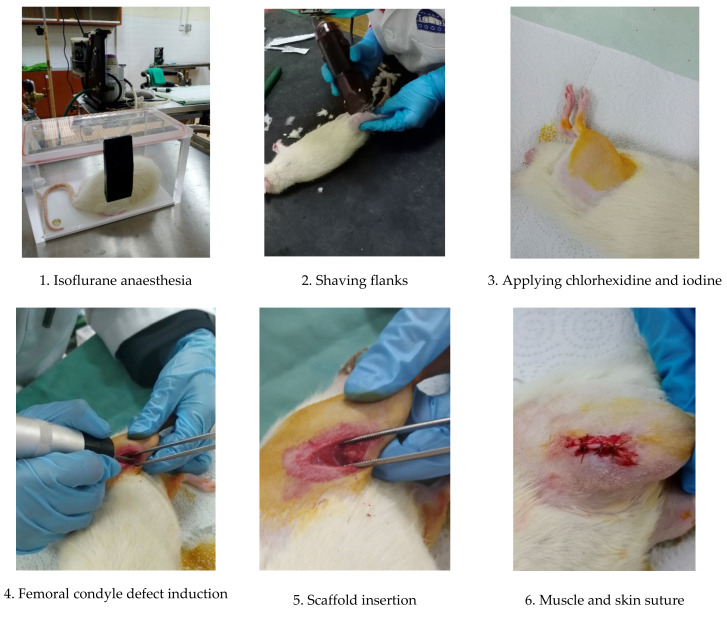
A summary of the scaffold implantation procedures.

**Table 1 pharmaceuticals-16-00876-t001:** The macroscopic monitoring for the harvested condyles. Note: R1-R9 means Rats 1–9. R means right condyle and L means left condyle. The Kruskal–Wallis test showed *p* > 0.05 for the investigated criteria, possibly due to the small sample size used in this preliminary study.

Samples																		
	**R1L**	**R1R**	**R2L**	**R2R**	**R3L**	**R3R**	**R4L**	**R4R**	**R5L**	**R5R**	**R6L**	**R6R**	**R7L**	**R7R**	**R8L**	**R8R**	**R9L**	**R9R**
Total score for each sample	5	3	5	5	3	3	6	3	5	3	7	3	5	3	7	6	3	6
Standard deviation	0.58	0	0.58	0.58	0	0	0	0	0.58	0	0.58	0	0.58	0	0.58	0	0	1

**Table 2 pharmaceuticals-16-00876-t002:** Scaffold compositions for the in vivo implantation.

Sample ID	Ratio HAp:FAp	Weight (g)			Volume (µL)			Volume (mL)
		(MW) CS	HAp	FAp	BP	PEG600 DMA	5 mg/mL P28	Acetic Acid
12% LW CS/HAp/FAp 1:1	1:1	(LW) 1.5	0.75	0.75	5	100	0	12.5
12% HW CS/HAp/FAp 1:1	1:1	(HW) 1.5	0.75	0.75	5	100	0	12.5
12% HW CS/HAp/FAp 1:0.75	1:0.75	(HW) 1.5	1	0.75	5	100	0	12.5
12% HW CS/HAp/FAp 1:1/P28 25 µg	1:1	(HW) 1.5	0.75	0.75	5	100	5	12.5
12% HW CS/HAp/FAp 1:1/P28 75 µg	1:1	(HW) 1.5	0.75	0.75	5	100	15	12.5
12% HW CS/HAp/FAp 1:1/P28 150 µg	1:1	(HW) 1.5	0.75	0.75	5	100	30	12.5

**Table 3 pharmaceuticals-16-00876-t003:** The randomised implantation table outlines the scaffold formulations implanted into each condyle of the nine rats.

Rat Tag	Left Condyle Treatments	Right Condyle Treatments
R1	12% LW CS/HAp/FAp 1:1	12% HW CS/HAp/FAp 1:1
R2	12% LW CS/HAp/FAp 1:1	12% HW CS/HAp/FAp 1:0.75
R3	12% LW CS/HAp/FAp 1:1	12% HW CS/HAp/FAp 1:1/P28 25 µg
R4	12% HW CS/HAp/FAp 1:1	12% HW CS/HAp/FAp 1:1/P28 25 µg
R5	12% HW CS/HAp/FAp 1:0.75	12% HW CS/HAp/FAp 1:1/P28 75 µg
R6	12% HW CS/HAp/FAp 1:1/P28 25 µg	12% HW CS/HAp/FAp 1:1/P28 75 µg
R7	12% HW CS/HAp/FAp 1:1/P28 150 µg	12% HW CS/HAp/FAp 1:0.75
R8	12% HW CS/HAp/FAp 1:1/P28 150 µg	12% HW CS/HAp/FAp 1:1
R9	12% HW CS/HAp/FAp 1:1/P28 150 µg	12% HW CS/HAp/FAp 1:1/P28 75 µg

**Table 4 pharmaceuticals-16-00876-t004:** Modified histopathological evaluation scoring (Rudert et al., 2005).

Criterion	Score	Macroscopic Characteristics
Defect visibility	1	Appeared as small irregular bumps
	2	Regular hole closed by a transparent tissue
	3	Completely closed
Colour	1	Yellowish paste observed
	2	White bony appearance
Surface	1	Rough and bumpy
	2	Smooth

## Data Availability

The data presented in this study are available on request from the corresponding author. The data are not publicly available due to ethical restrictions.

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
