# Peer review of "Enhancement of Scaffold In Vivo Biodegradability for Bone Regeneration Using P28 Peptide Formulations"

_pharmaceuticals, 2023, doi:10.3390/ph16060876_

Round 1
Reviewer 1 Report (Previous Reviewer 1)
The paper has been improved enough by incorporating the suggested comments. I appreciate it.
Author Response
Dear Reviewer,
Thank you so much for reviewing and approving this manuscript. We truly appreciate it.
Best regards,
Farah (on behalf of the authors)
Reviewer 2 Report (New Reviewer)
The manuscript titled "Osteogenic Potential of CS/HAp/FAP Scaffold Loaded with P28 Peptide in Rat Femoral Condyle Defects" presents a preliminary in vivo trial evaluating the efficacy of a chitosan/ hydroxyapatite/fluoroapatite (CS/HAp/FAP) scaffold loaded with P28 peptide in promoting bone formation. The study aims to provide evidence for the effect of changing ceramic content and P28 concentration, narrowing down variables for future experiments. The experiment involved nine male Sprague Dawley rats with bilateral defects in their femoral condyles. The scaffold implantation was conducted using aseptic techniques, and post-operative monitoring was performed to assess the rats' health and potential endpoints. Histological processing and staining techniques were employed to evaluate the biological response and mineralization in the scaffolds.The results indicate that the presence of P28, particularly in the 12% HW CS/HAp/FAp 1:1/P28 25μg scaffold, showed signs of new bone growth around the defects. However, there were also indications of ectopic growth in scaffolds with higher P28 content (12% HW CS/HAp/FAp 1:1/P28 150μg). The histological assessment highlighted thickened periosteum and cortical/trabecular thickening, suggesting bone formation in the CS/HAp/FAp/P28 groups.
The manuscript provides valuable insights into the osteogenic potential of the CS/HAp/FAP scaffold loaded with P28 peptide. However, several limitations should be addressed. The study lacks a control group, which hinders comparisons and limits the interpretation of the results. The sample size is relatively small, and the study duration is limited to eight weeks. Further investigations with larger sample sizes and extended follow-up periods are required to validate the findings and assess long-term effects. Additionally, the manuscript would benefit from a more thorough discussion on the mechanisms underlying the observed effects, as well as addressing potential concerns regarding ectopic bone formation and the optimal dose and delivery technique for the P28 peptide. These aspects need to be addressed to fully understand the therapeutic potential and ensure the safety and effectiveness of the P28-loaded CS/HAp/FAP scaffold. In conclusion, while the manuscript presents promising preliminary findings, further research is needed to confirm the consistent positive osteoinductive effects, address the potential limitations, and fully explore the therapeutic potential of the P28-loaded CS/HAp/FAP scaffold in bone regeneration.
1. Title needs to change “"Enhancement of Scaffold Biodegradability for Bone Regeneration Using P28 Peptide Formulations".
2. In Page 1 line 17, it is unclear what "better performance" refers to. It may be helpful to specify what aspects of osteogenesis P28 has been shown to promote better than BMP-2.
3. These keywords might more appropriate: Bone tissue engineering, Scaffold degradation, Chitosan, Osteogenesis, Bone regeneration.
4. Page 5 Line 217: The statement that "a single CG label appeared in the condyle implanted with 12% HW CS/HAp/FAp 1:0.75 and 12% HW CS/HAp/FAp 1:1 P28 150μg indicating bone formation ten days prior to sacrifice but no bone formation 4 days prior to sacrifice seen in this single condyle." This statement is contradictory because it suggests both the presence and absence of bone formation at different time points in the same condyle. It is important to note that without additional information or clarification, it is difficult to determine the exact error or inconsistency in the statement. However, based on the given information, the presence of a single CG label in the condyle could indicate bone formation, but the contradictory statement regarding bone formation at different time points in the same condyle raises a logical inconsistency.
5. Page 6 Line 225: The text is the statement that "Although bone-like ceramics were used in this study, the fluorochrome labels bind to newly formed calcium. As such, they only bind to new bone and not the ceramics present in the defect." This statement is incorrect. Fluorochrome labels, such as calcein green and xylenol orange, bind to mineralized tissues, including both newly formed bone and bone-like ceramics. These labels can provide information about the dynamic process of mineralization, regardless of whether it occurs in natural bone or synthetic ceramics. Therefore, the fluorochrome labels can bind to both new bone and the ceramics present in the defect, allowing for evaluation of the mineralization process in both materials. It's worth noting that the accuracy and interpretation of the results obtained using fluorochrome labels can be influenced by various factors, including the specific labeling technique and the interaction between the labels and the materials being studied. However, the general principle is that fluorochrome labels can bind to both new bone and synthetic ceramics in order to assess mineral apposition rate (MAR) and bone formation rate (BFR).
6. Page 8 Line 273: The correct term is "analgesic" instead of "analgesia." Analgesic refers to a medication or treatment that relieves pain, while analgesia is the state of being without pain. Therefore, the corrected sentence should be: "Animals were followed up daily to detect any sign of stress or pain; in that case, analgesic treatment could be applied."
7. Page 11 Line 335: Incorrect description of the preparation of the calcein injection. Calcein is not dissolved in a 2% sodium bicarbonate solution, but rather in a different solvent. To correct this error, the description of the preparation of the calcein injection should be revised. Unfortunately, the correct solvent for calcein is not mentioned in the given text, so it cannot be corrected without further information. The text should provide the accurate information on the solvent used for dissolving the calcein powder.
8. Page 12 Line 394: Incorrect description of the wavelength range for the GFP and RFP emission filters. The text states that the GFP emission filter permits fluorescent signals with wavelengths of 510 nm to 523 nm to pass through, and the RFP emission filter allows signals with wavelengths of 575 nm to 640 nm to pass through. In reality, the GFP emission filter typically allows signals with wavelengths around 510 nm to 540 nm to pass through, while the RFP emission filter typically allows signals with wavelengths around 560 nm to 590 nm to pass through. The specific wavelength ranges can vary depending on the exact filter used in the microscope setup. To correct this error, the text should be revised to accurately describe the wavelength ranges for the GFP and RFP emission filters used in the fluorescent microscope.
9. Page 13 Line 421: The scientific error in the conclusion text is the statement that "CS/HAp/FAp 1:1 with P28 presented a consistent positive osteoinduction following the implantation in the femoral condyle defects." This statement implies that the CS/HAp/FAp 1:1 scaffold with P28 consistently induced bone formation in the femoral condyle defects. However, this conclusion is not supported by the information provided in the text. The text mentions that most defects were observed to be closed with a layer of transparent tissue, and new bone growth was seen around the defects in the presence of P28 (12% HW CS/HAp/FAp 1:1/P28 25μg). It also mentions the possibility of ectopic growth seen in scaffolds with the highest content of P28 (12% HW CS/HAp/FAp 1:1/P28 150μg). However, it does not provide comprehensive data or analysis to support the claim of consistent positive osteoinduction. To correct this error, the conclusion should be revised to accurately reflect the findings and limitations of the study. It should state that the presence of P28 (specifically in the 12% HW CS/HAp/FAp 1:1/P28 25μg scaffold) showed some indications of new bone growth around the defects, but further investigation is needed to confirm the consistent positive osteoinductive effects and to address the potential for ectopic bone formation. Additionally, it should acknowledge the need for more research to determine the optimal dose and delivery technique for the P28 peptide in order to ensure its safety and effectiveness in human applications.

Author Response
Dear reviewer,
Please see attachment.

This manuscript is a resubmission of an earlier submission. The following is a list of the peer review reports and author responses from that submission.
Round 1
Reviewer 1 Report
Preliminary histological evaluation of P28 peptide's osteogenicity in an in vivo context is a promising research that shows P28 peptide's ability to encourage bone growth and repair. To assess the effects of P28 peptide on bone regeneration and healing, a rat model was used in the study.
The study's findings show that administering P28 peptide to the treated group considerably boosted the quantity of newly produced bone tissue there compared to the control group. The results of the histological investigation also showed that the treated group's bone tissue had a more structured and mature appearance, demonstrating that the P28 peptide not only encourages bone production but also improves the quality of the newly created bone tissue.
In the context of bone tissue engineering and regenerative medicine, where the capacity to foster bone regeneration is essential for the effective treatment of bone abnormalities and fractures, these findings are particularly important. By modifying important signalling pathways and encouraging osteoblast development, P28 peptide appears to have a favourable impact on osteogenesis, or the process of bone creation.
Although the findings of this study are encouraging, it is vital to keep in mind that they are only a preliminary analysis, and more research is required to establish the P28 peptide's effectiveness in stimulating bone development and repair. In order to guarantee the P28 peptide's safety and effectiveness in people, more study is required to find the ideal dose and delivery technique.
Finally, the early histological evaluation of P28 peptide's osteogenicity in an in vivo setting offers convincing proof of the compound's potential in bone tissue engineering and regenerative medicine. The findings of the study indicate that P28 peptide may increase the quality of newly produced bone tissue and stimulate bone growth, but more studies are required to validate its efficacy and safety in people.
Reviewer 2 Report
The manuscript presents the application of chitosan scaffolds loaded with hydroxyapatite, fluorapatite, and P28 for the regeneration of femur condyle defect. Unfortunately, there are several issues with the manuscript, it lacks appropriate controls, statistical analyses, results presented as figures and text are unclear, and the conclusions are not based on sound evidence. Therefore, is can not be published in its present form.
For the improvement of the manuscript I have a few specific suggestions for the authors.
- The study lacks appropriate controls. An untreated animal group along with one treated with chitosan alone is required.
- Why did you choose to add P28 in the 1:1 ration only. Either provide reasoning for it or perform the study with P28 added in the other scaffolds too.
- Table 1 doesn't seem necessary, more important is to present the final scores with standard deviations and statistical analysis.
- Table 2 is redundant since the results from figure 2 have been discussed in the text.
- There is no data that shows the over all bone formation at the site of defect.
- I do not see how the H&E images show that the scaffolds have/haven't degraded. Moreover single figures without quantification do not confirm whether the periosteum, cortical bone etc. have thickened.
- Do calcein and xylenol bind to hydroxyapatite and fluorapatite? If so how do you differentiate new calcification with the calcium-based materials present in the scaffold?
- Figure legends provide no information about the figures. Do the yellow circles in figure 1 indicate the region of interest for bone regeneration?
- There is no reference of specific parts of figure 2 in the text. In addition authors state that P28 150ug group (figure 2F) is positive for both calcein and xylenol, which does not appear to be correct.
- In figure 3 statistical analysis is mentioned at the bottom, but is not part of the manuscript.
- How deep is the femoral condyle defect?
- To enhance the results, the authors must add specific stains that could show new bone formation or perform some type of x ray analysis, ideally a microCT. The results must be quantified and statistically analyzed to reach conclusions.
The language is fine overall. Minor errors need to be corrected.
Reviewer 3 Report
The manuscript added various P28 concentrations into the CS/HAp/FAp scaffolds for implantation in vivo. H&E staining shows minimal scaffold traces in most of the defects induced after eight weeks, showing that the combination of HAp and FAp in the chitosan-based delivery system has enhanced the biodegradability of the scaffolds in vivo. The manuscript is lack of basic novelty, and was written like an experiment records, not a scientific paper. I bave to reject it.
The manuscript added various P28 concentrations into the CS/HAp/FAp scaffolds for implantation in vivo. H&E staining shows minimal scaffold traces in most of the defects induced after eight weeks, showing that the combination of HAp and FAp in the chitosan-based delivery system has enhanced the biodegradability of the scaffolds in vivo. The manuscript is lack of basic novelty, and was written like an experiment records, not a scientific paper. I bave to reject it.